# Training-time Selection of Linear Vs. Softmax Attention in Layer-based Hybrid Transformers

## Abstract

Given a prompt of initial length $M$ to generate $N$ tokens, the current transformer-based LLMs' memory requirement grows with $\mathcal{O}(M + N)$, while the inference time grows with $\mathcal{O}(MN + N^2)$. While these models have achieved remarkable results across various tasks, their rapid growth rates set upper limits for enhancing the accuracy of the output via increasing the context length due to memory size constraints and time requirements. Linear attention mechanisms (Peng et al., 2021) offer $\mathcal{O}(N)$ time and constant memory complexity but fall short in some language modeling tasks, especially when the softmax-attention is using the full context. A natural direction is to design hybrid models that combine the strengths of both approaches. In this work, we propose a training-based method for constructing such hybrid models. This method aims to find the optimal layer-based hybrid configuration of a transformer given a maximum tolerable incremental loss. The method aims to replace any softmax attention block with its linear counterpart, so long as it does not incur additional loss beyond a desirable tolerable limit. We evaluate our hybrid models on various language-modeling benchmarks. The result shows that the hybrid models obtained by this method, in some cases, can cut the LLM's total context cache peak memory usage by up to 40 % while affecting the accuracy minimally (increasing perplexity by 1%). Furthermore, we observe that our method of training, in some cases, even results in a reduction of the task-specific loss (e.g., cross-entropy) compared to an all softmax-attention configuration. Therefore, using the proposed method not only makes the model more efficient in terms of memory usage and compute intensity but also increases the accuracy, i.e., reduces perplexity. We show that early and final layers can usually be replaced with linear attention layers, while the mid layers must preserve softmax attention, and the exact pattern differs from dataset to dataset.

## 1 Introduction

Since their introduction, transformers have become the foundation of state-of-the-art language models (Vaswani et al., 2017). In the context of autoregressive language modeling, a Transformer primarily consists of two key components: Self-Attention and Feed-Forward Network (FFN). The self-attention mechanism is defined as:

$$\text{Attention}(Q, K, V) = \text{softmax}\Big(\frac{QK^\top}{\sqrt{d_k}} + \text{CausalMask}\Big) V$$

where $Q$, $K$, and $V$ are the query, key, and value matrices obtained via learned linear projections of the hidden states. In practice, the hidden states $H$ from the previous layer are projected as $Q = HW_Q$, $K = HW_K$, $V = HW_V$, with $W_Q, W_K, W_V \in \mathbb{R}^{d \times d_k}$, where $d_k$ is the key vector dimension.

### 1.1 Self-Attention Mechanism and KV-Cache

The first time an LLM processes a user's prompt is referred to as the **prefill stage**. In this stage, all $M$ input tokens are processed in parallel through the Transformer stack, and the key-value cache (KV-cache) is computed and stored for each layer. Formally, let $X \in \mathbb{R}^{M \times d}$ denote the input embedding

matrix (with $M$ the prompt length and $d$ the model dimension). $X$ is obtained after the embedding stage and applying the positional encodings, and is ready to be passed to the first transformer. $X$ is passed through $L$ Transformer layers, producing hidden states $H^{(i)}$ at layer $i$ (with $H^{(0)} = X$), such that

$$H^{(i)} = \text{TransformerLayer}^{(i)}(H^{(i-1)}).$$

During the prefill, the self-attention at layer $i$ takes $Q^{(i)} = H^{(i-1)}W_Q$ and $K^{(i)} = H^{(i-1)}W_K$, $V^{(i)} = H^{(i-1)}W_V$ for all $M$ tokens. The resulting $K^{(i)}$ and $V^{(i)}$ ($\in \mathbb{R}^{M \times n_{\text{head}} \times d_k}$ for $n_{\text{head}}$ attention heads and $d_k = d/n_{\text{head}}$) are stored in the KV-cache for that layer:

$$\text{cache}^{(i)} = (K^{(i)}, V^{(i)}),$$

with the full cache consisting of $\{\text{cache}^{(i)} : i = 1, \ldots, L\}$.

Once the KV-cache is initialized for all layers, the model enters the **decoding stage**. In decoding (generation), the model produces one token at a time autoregressively. Let $x_t \in \mathbb{R}^{1 \times d}$ be the embedding of the newly generated token at step $t$. This $x_t$ is fed to the Transformer layers. At layer $i$, the query $q_t^{(i)} = H_t^{(i-1)}W_Q$ (a $1 \times d_k$ vector) attends to the keys and values of all previous tokens stored in the cache:

$$\text{Attention}\big(q_t^{(i)}, K_{1:t}^{(i)}, V_{1:t}^{(i)}\big) = \text{softmax}\bigg(\frac{q_t^{(i)} K_{1:t}^{(i)\top}}{\sqrt{d_k}}\bigg) V_{1:t}^{(i)} \tag{1}$$

where $K_{1:t}^{(i)}$ and $V_{1:t}^{(i)}$ are the concatenation of the cached keys and values from all past $t$ tokens (including the new token). After computing the attention output, the new key $k_t^{(i)}$ and value $v_t^{(i)}$ are appended to the cache for layer $i$. This process repeats for $t = 1$ to $N$ to generate $N$ tokens.

**Complexity Analysis.** Suppose the model has seen a prompt of length $M$ and we aim to generate $N$ additional tokens. *Memory complexity:* storing the KV-cache for all $M + N$ tokens across all layers is $\mathcal{O}(M + N)$ (linear in sequence length). *Time complexity:* each decoding step attends to all prior tokens, leading to $\sum_{t=1}^{N}(M + t) = MN + \frac{N(N+1)}{2} = \mathcal{O}(MN + N^2)$.

## 2 MOTIVATION

Recent state-of-the-art LLMs can handle tens or even hundreds of thousands of tokens in a single prompt; for instance, LLama 4 Scout has a context window of size 10 million token (Meta, 2025). The longer context length enables the user to summarize large chunks of text corpus; perhaps entire books. Study has shown that enabling transformers to capture longer contexts can significantly improve the Language Model's accuracy on downstream tasks (Zhang et al., 2023).

However, these advantages come at a steep memory cost. The self-attention mechanism's $\mathcal{O}(N^2)$ time and space complexity creates a serious *memory wall* as $N$ grows. Handling extremely long sequences is both memory-intensive and slow. For instance, Llama-2 7B in BF16 required 14GB of memory for model weights, while KV cache adds 69 GB to this number, which exceeds H100 GPU capacity (Ge et al. (2025)).

Therefore, it is critically important to develop faster and more memory-efficient alternatives to standard self-attention for long sequences.

## 3 RELATED WORKS

Recent works have used various KV-caching techniques to tackle the problem. The main theme of these works is selecting the tokens that have the greatest effect on the output of the model. Streaming LLM (Xiao et al., 2024), for instance always keeps a rolling window of the recent tokens alongside a few initial tokens of the text. Another work (Ge et al., 2025) uses a hybrid approach where the layers in the middle of the model use a reduced KV-cache while the other layers use the full context. Although, these methods show a reduction in KV-cache memory usage, their usage depends on choosing a number of hyperparameters such as the reduced layers, window size, number of initial

tokens, and etc. The optimal choice of these hyperparameters might vastly differ for different tasks and different models. Finding the optimal choices often requires exhaustive search and may not be practical.

Another direction is to use other alternatives rather than softmax-attention. For instance, a variety of linear attention mechanisms have been proposed . One of the early examples was Peng et al. (2021). Linear attention applies a non-linear kernel on query and key vectors so that it can replace softmax attention:

$$\text{LinearAttention}\big(q_t, \{k_i\}_{i=1}^t, \{v_i\}_{i=1}^t\big) = \sum_{i=1}^t \frac{\phi(q_t)\phi(k_i)^\top v_i}{\sum_{j=1}^t \phi(q_t)\phi(k_j)^\top} = \frac{\phi(q_t)S_t}{\phi(q_t)z_t^\top} \qquad (2)$$

Where $S_t$ and $z_t$ are defined as follows:

$$S_t = \sum_{i=1}^t \phi(k_i)^\top v_i \qquad z_t = \sum_{i=1}^t \phi(k_i)$$

and $\phi(.)$ is non-linear mapping such as ReLU (Kasai et al., 2021), elu (Katharopoulos et al., 2020), random transformations (Peng et al., 2021) and etc. Therefore, linear attention effectively eliminates the need for keeping a growing KV-cache in memory by keeping the while context in two fixed-size matrices $S_t$ and $z_t$. Although state-of-the-art linear attention (with no gating mechanism) takes much smaller memory than softmax-attention, it still suffers from a gap in accuracy (Zhang et al., 2024). The addition of gating mechanism to the linear attention block has improved the quality of linear models significantly (Yang et al., 2024). Yang et al. (2025), for instance, has outperformed other linear variants as well as conventional transformer.

In this work, we present a hybridization method which given a particular linear attention variant, finds an optimal layer allocation. This can be categorized as a neural architecture exploration using Straight-through Gumbel-softmax which tries to minimize the usage of softmax-attention block without harming the accuracy beyond a tolerable limit. Usage of Gumbel-softmax for architecture exploration has been studied across different tasks such as image classification, language modeling, and recommender models (Chang et al., 2019; Zeng et al., 2023; Chen et al., 2021). Here we use this method for choosing linear attention throughout the model wherever it can be done without going beyond a tolerable loss.

## 4 METHODOLOGY

Here we propose a training-based layer-based hybridization method for achieving more efficient LLMs while trying to preserve the same level of accuracy that can work across different datasets. At training time, we introduce two parallel attention blocks namely linear and softmax attention. The final goal is to decide which layers' conventional attention blocks can be replaced by the more efficient linear attention. Various kernel functions for the linear attention block have been proposed such as $1 + \text{elu}(\cdot)$ (Katharopoulos et al. (2020)) and linear transformation with ReLU (Kasai et al. (2021)). In this work, we use a normalized $\exp(\cdot)$ function used in Lu et al. (2025). Algorithm 1 shows the overall process of training.

$$\text{SamplerLogit}^{(j)} := s^{(j)} \in \mathbb{R} \qquad j = 1, \dots, L$$

To map SamplerLogit into a bounded value, a scaled $\tanh(\cdot)$ function is applied to produce DecisionLogit for each layer:

$$\text{DecisionLogit}^{(j)} := d^{(j)} = K \tanh(s^{(j)}) \qquad j = 1, \dots, L$$

Where $K$ is a positive real hyperparameter. Using DecisionLogit values, we construct the softmax-attention and linear-attention gates, respectively:

$$(g_s^{(j)}, g_l^{(j)}) := (-\tfrac{1}{2}d^{(j)}, \tfrac{1}{2}d^{(j)}) \quad j = 1, \dots, L$$

At iteration $t$, we use the temperature $\tau_t$ to generate the probabilities by which one of the attention blocks is chosen:

$$(p_{s,t}^{(j)}, p_{l,t}^{(j)}) = \operatorname{softmax}\left(\tfrac{1}{\tau_t}(g_s^{(j)}, g_l^{(j)})\right) \qquad j = 1, \dots L$$

Our goal is to learn this probability distribution for each layer so that after training, we can replace the layers with $p_{s,T}^{(j)} < p_{l,T}^{(j)}$ where $T$ denotes the last iteration of training, with linear attention (or equivalently $s^{(j)} > 0$). This means that the network has learned that it is possible to use linear attention for this layer most of the times. We later show how the training method promotes the use of linear attention but for now we are concerned with a method to learn the $s^{(j)}$ values, thus learning the $(p_{s,t}^{(j)}, p_{l,t}^{(j)})$ probability distribution for all layers through backpropagation. More formally, we want to sample/choose a softmax attention block according to $(p_{s,t}^{(j)}, p_{l,t}^{(j)})$ which is a continous distirbution on the simplex. The idea is that after numerous iterations of training and with different batches, the model learns to push the probabilites assigned to each block in a way that the suitable attention-block will have a larger probability of being sampled. We anneal the temperature as training goes forward to make the model converge to a one-hot probability distribution. The choice of the attention block for layer $j$ can be modeled with a single random variable:

$$y_{s,t}^{(j)} \sim \operatorname{Bernoulli}(p_{s,t}^{(j)})$$

Thus, the output of attention at iteration $t$ of the training process would be:

$$\operatorname{AttnOut}^{(j)}(x; t) = y_{s,t}^{(j)}.\operatorname{SoftmaxAttn}^{(j)}(x) + (1 - y_{s,t}^{(j)}).\operatorname{LinearAttn}^{(j)}(x) \tag{3}$$

However, the gradient estimation of a sampling process from a categorical distribution is not trivial. Fortunately, there has been extensive research on this topic and one reparametrization technique called Gumbel-Softmax can be used. Gumbel-Softmax is a differentiable mechanism that allows the model to learn categorical variables via standard backpropagation (Jang et al., 2017) and (Maddison et al., 2017). Particularly, we use Straight-through Gumbel-Softmax (ST-Gumbel-Softmax) to produce $y_{s,t}^{(j)}$ given $t$ and $(g_s^{(j)}, g_l^{(j)})$. Straight-through property ensures that in forward pass $y_{s,t}^{(j)}$ is either one or zero (only one blocked used and the two attentions block's output will not be averaged) while the backward pass pretends that the output was a soft Gumbel-Softmax so the graidents can flow backwards and update both linear and attention blocks at each iteration $t$. Therefore, we have:

$$y_t^{(j)} = (y_{s,t}^{(j)}, y_{l,t}^{(j)}) = \text{ST-Gumbel-Softmax}(g_s^{(j)}, g_l^{(j)}; \tau_t) \tag{4}$$

Furthermore, we need to modify the loss function to promote the use of more efficient linear layers when possible. First, we introduce $\mathcal{L}_{SM}$ which is the sum of layers that have been chosen to be softmax throughout the model for the current batch:

$$\mathcal{L}_{\text{SM}}(y_t) = \sum_{j=1}^{L} y_{s,t}^{(j)}$$

where $y_t = (y_t^{(1)}, y_t^{(2)}, \cdots, y_t^{(L)})$. If minimizing $\mathcal{L}_{\text{task}}$ is the primary objective of the training e.g. cross-entropy loss in language pretraining, then, the modified task loss would be defined as:

$$\mathcal{L}_{\text{total}}(x; \psi, y_t, t) = \mathcal{L}_{\text{task}}(x; \psi, y_t) + \frac{\lambda}{L}\mathcal{L}_{\text{SM}}(y_t) = \mathcal{L}_{\text{task}}(x; \psi, y_t) + \frac{\lambda}{L}\sum_{j=1}^{L} y_{s,t}^{(j)} \tag{5}$$

Where $\psi$ denotes the network parameters and $x$ is the input. In this framework, $\lambda$ controls the trade-off between efficiency and accuracy. Higher values of $\lambda$ will promote a smaller number of softmax blocks selected by the training method to decrease the total loss while minimizing $\frac{\lambda}{L}\mathcal{L}_{SM}$. Under certain conditions, the model will replace a softmax-attention with its linear counterpart if and only if this replacement will not add any incremental loss higher than $\frac{\lambda}{L}$ to the optimal loss, which would be otherwise obtained by a conventional all-softmax training setting. Therefore, in case, all layers are replaced by their linear counterpart during training, the additional loss to the optimal loss would not be greater than $\lambda$. Thus, we can consider $\lambda$ to be the maximum tolerable incremental loss.

After training, for each layer, for the layers with positive $s^{(j)}$, we'll choose the trained linear attention while choosing the trained softmax attention for the other layers.

---

**Algorithm 1** Training Algorithm

---

1: Initialize $s^{(j)} \leftarrow 0 \quad j = 1, 2, ..., L$
2: Initialize $\psi$ (model Parameters) randomly.
3: Initialize $\lambda$ to the maximum tolerable incremental loss and $K$ to a positive real number
4: Choose a schedule for temperatures: $\tau_t = \tau(t)$
5: **for** $t = 1$ to $B$ **do**
6: $\quad (x, h) \leftarrow \text{Batch}_t$
7: $\quad \mathcal{L}_{\text{SM}}(y_t) = 0$
8: $\quad x^{(0)} = \text{Embedding}(x)$
9: $\quad$ **for** $j = 1$ to $L$ **do**
10: $\quad\quad \tau_t \leftarrow \tau(t)$
11: $\quad\quad (y_{s,t}^{(j)}, y_{l,t}^{(j)}) \leftarrow \text{ST-Gumbel-Softmax}(-\frac{K}{2}\tanh(s^{(j)}), \frac{K}{2}\tanh(s^{(j)}); \tau_t)$
12: $\quad\quad a^{(j)} \leftarrow \text{LayerNorm}(x^{(j-1)})$
13: $\quad\quad x^{(j)} \leftarrow x^{(j-1)} + y_{s,t}^{(j)}.\text{SoftmaxAttn}^{(j)}(a^{(j)}) + (1 - y_{s,t}^{(j)}).\text{LinearAttn}^{(j)}(a^{(j)})$
14: $\quad\quad x^{(j)} \leftarrow x^{(j)} + \text{FFN}(\text{LayerNorm}(x^{(j)}))$
15: $\quad\quad \mathcal{L}_{\text{SM}}(y_t) \leftarrow \mathcal{L}_{\text{SM}}(y_t) + y_{s,t}^{(j)}$
16: $\quad$ **end for**
17: $\quad \hat{h} = \text{DeEmbedding}(x^{(L)})$
18: $\quad$ Calculate $\mathcal{L}_{task}(x, h, \hat{h}; \psi, y_t)$
19: $\quad$ Calculate $\mathcal{L}_{total}(x, h, \hat{h}; \psi, y_t, t) = \mathcal{L}_{task}(x, h, \hat{h}; \psi, y_t) + \frac{\lambda}{L}\mathcal{L}_{\text{SM}}(y_t)$
20: $\quad$ Backpropagate and Update $\psi, s^{(j)} \quad j = 1, 2, ..., L$
21: **end for**
22: **return** $\psi, s^{(j)} \quad j = 1, 2, ..., L$

---

## 5 CONVERGENCE WITH ANNEALED TEMPERATURE

**Model and parametrization.** For each layer $j \in \{1, \ldots, L\}$ we store a scalar $s^{(j)} \in \mathbb{R}$ and define the bounded logit gap

$$d^{(j)} = K \tanh(s^{(j)}) \in [-K, K] \quad (K > 0).$$

Centered logits are

$$(g_s^{(j)}, g_l^{(j)}) = \left(-\tfrac{1}{2}d^{(j)}, \tfrac{1}{2}d^{(j)}\right).$$

At iteration $t$ with temperature $\tau_t > 0$ the soft gate is

$$p_t^{(j)} = (p_{s,t}^{(j)}, p_{l,t}^{(j)}) = \text{softmax}\left(\tfrac{1}{\tau_t}(g_s^{(j)}, g_l^{(j)})\right)$$

Hard one-hot gates $y_t^{(j)}$ are drawn by the Gumbel-Max trick the soft relaxation follows the Gumbel-Softmax/Concrete distributions (Jang et al., 2017; Maddison et al., 2017). Training uses the straight-through (ST) estimator (Bengio et al., 2013). The forward uses $y_t^{(j)}$, while gradients follow $p_t^{(j)}$.

**Losses and population objective.** For a minibatch $\xi = (x, t)$ and network output $f(x; \psi, y_t)$ computed with hard gates $y_t = (y_t^{(1)}, \ldots, y_t^{(L)})$, $s = (s^{(1)}, \ldots, s^{(L)})$, and $p_t = (p_t^{(1)}, \ldots, p_t^{(L)})$the per-step loss and the expected objective at temperature $\tau$ are

$$\ell(\psi, s; \xi, y_t) = \text{CE}(f(x; \psi, y_t), t) + \frac{\lambda}{L}\sum_{j=1}^{L} y_{s,t}^{(j)} \tag{6}$$

$$\bar{J}_t(\psi, s) = \mathbb{E}_{\xi,y}\left[\text{CE}(f(x; \psi, y_t), t)\right] + \frac{\lambda}{L}\sum_{j=1}^{L} p_{s,t}^{(j)} \tag{7}$$

**Lemma 1** (Hard/soft equivalence of the auxiliary term)**.** *For each $i$, any $\tau > 0$ and any $t$,* $\mathbb{E}[y_{s,t}^{(j)} \mid s^{(j)}, \tau_t] = p_{s,t}^{(j)}$. *Hence* $\mathbb{E}[\frac{\lambda}{L}\sum_i y_{s,t}^{(j)}] = \frac{\lambda}{L}\sum_j p_{s,t}^{(j)}$.

**Lemma 2** (Zero-temperature limit of the soft gate)**.** *For any fixed $p_t^{(j)}$ with $d^{(j)} \neq 0$,*

$$\lim_{\tau \downarrow 0} p_{l,t}^{(j)} = \mathbf{1}\{d^{(j)} > 0\}, \qquad \lim_{\tau \downarrow 0} p_{s,t}^{(j)} = \mathbf{1}\{d^{(j)} < 0\}.$$

**Algorithm (two-time-scale stochastic approximation).** With stepsizes $\eta_t > 0$ and a decreasing schedule $\tau_t \downarrow 0$, one update is

$$(\psi_{t+1}, s_{t+1}) = (\psi_t, s_t) - \eta_t \widehat{\nabla} \bar{J}_{\tau_t}(\psi_t, s_t; \xi_t, y_t), \tag{8}$$

where $\widehat{\nabla} \bar{J}_{\tau_t}$ is the ST gradient at temperature $\tau_t$ (hard forward, soft backward).

**Assumptions.**

**A1** (*Stepsizes*) $\eta_t > 0$, $\sum_t \eta_t = \infty$, and $\sum_t \eta_t^2 < \infty$.

**A2** (*Annealing*) $\tau_t \downarrow 0$ with $\dfrac{\eta_t}{\tau_t} \to 0$ and $\sum_t \eta_t |\tau_{t+1} - \tau_t| < \infty$. (A sufficient choice is $\eta_t = \eta_0/(t+1)^\alpha$, $\tau_t = \tau_0/(t+1)^\beta$ with $\alpha \in (1/2, 1]$, $\beta \in (0, \alpha)$.)

**A3** (*Regularity, uniform in $\tau$*) For $\tau \in (0, \tau_0]$, the mapping $\theta \mapsto \nabla \bar{J}_\tau(\theta)$ (with $\theta = (\phi, s)$) is locally Lipschitz; the per-step ST gradient has a uniformly bounded second moment: $\sup_{\tau \in (0,\tau_0]} \mathbb{E}[\|\widehat{\nabla}\bar{J}_\tau(\theta_t)\|^2 \mid \theta_t] \leq C(1 + \|\theta_t\|^2)$.

**A4** (*ST as pseudogradient, uniform in $\tau$*) There exists $c > 0$ such that for all $\tau \in (0, \tau_0]$,
$$\mathbb{E}[\widehat{\nabla}\bar{J}_\tau(\theta) \mid \theta] \cdot \nabla \bar{J}_\tau(\theta) \geq c \|\nabla \bar{J}_\tau(\theta)\|^2.$$

**A5** (*Compact effective domain*) Because $d^{(j)} \in [-K, K]$ for all $j$, the logits $[g_s^{(j)}, g_l^{(j)}]$ lie in a compact set; sublevel sets of $\bar{J}_\tau$ are thus compact and gradients are bounded on them.

**Theorem 3** (Two-time-scale convergence with uniform $s$-penalty)**.** *Under A1–A5, the iterates* $(\psi_t, u_t)$ *of equation 8 satisfy:*

(i) ***Tracking of stationary sets.*** *Every limit point of $\{(\psi_t, s_t)\}$ lies in the upper semicontinuous limit of stationary sets $\Theta_\tau^\star := \{\nabla \bar{J}_\tau = 0\}$ as $\tau \downarrow 0$.*

(ii) ***One-hot gates almost surely.*** *For each layer $j$, $p_{l,t}^{(j)} \to 0$ or $1$ almost surely as $t \to \infty$ (the tie event $d^{(j)} = 0$ has probability $0$ due to sampling noise).*

(iii) ***Accuracy–efficiency selection rule.*** *Let $(\psi^\star, s^\star)$ be any limit point and let $\mathrm{CE}_j^i(\psi^\star)$ denote the contribution to the cross-entropy when block $j \in \{s, l\}$ is used in layer $i$ at parameters $\psi^\star$. For small $\tau$, layer $i$ selects the block*
$$\arg\min_{j \in \{s,l\}} \left\{ \mathrm{CE}_j^i(\psi^\star) + \tfrac{\lambda}{L} \mathbf{1}\{j = s\} \right\}.$$
*Equivalently, the layer switches to the efficient $l$-block iff*
$$\Delta \mathrm{CE}_i := \mathrm{CE}_l^i(\psi^\star) - \mathrm{CE}_s^i(\psi^\star) \leq \tfrac{\lambda}{L}.$$
*Thus only layers whose accuracy loss is at most $\lambda/L$ flip to $l$; the rest keep $s$.*

*Proof.* See Appendix A. □

## 6 EXPERIMENTS AND EVALUATION

The evaluation of this method can be categorized into 2 categories:

1. Language Modeling: These experiments aim to verify the effectiveness of the method in order to preserve the accuracy and language-modeling capabilities of the model after being trained with the proposed method. For this purpose, we have pretrained a GPT2-Style LLM (Radford et al., 2019) on three different datasets: Wikitext-103 (Merity et al., 2016), OpenWebText (Gokaslan et al., 2019), and PG19 (Rae et al., 2019). We have pretrained a baseline model with the original configuration alongside the modified model with 5 different $\lambda$ values. The cross-entropy loss and perplexity of them have been compared.

2. Measurement of KV-cache and total memory usage to show the runtime efficiency benefits of the proposed method at the time of inference

## 6.1 EXPERIMENT SETTING AND RESULTS

We use 4 NVIDIA H100 GPUs to train GPT-2 with the proposed training method with a batch size of 16 on Wikitext-103, PG19, and OpenWebText datasets. The original all-softmax attention transformer is trained with the same hyperparameters as the setting we used for hybrid model training. The training hyperparameter choices are detailed in Table 4. Several $\lambda$ values have been tested, namely: 0.05, 0.1, 0.25, 0.4, 0.55. Table 1 shows all the best validation losses for each penalty and each dataset. It is observed that for many of the penalties, the final validation loss will be lower than the original all-softmax baseline. As expected, for a fixed dataset, validation loss is an increasing function of $\lambda$. As explained in Section 5, the maximum tolerable loss is set by $\lambda$. In this table, we examine if, in fact, the training has satisfied this assumption. In all of the training settings, the incurred loss is less than $\lambda$.

Table 2 compares the perplexity achieved by fully linear LLMs compared to our work. The training reaches to a 0.5% of the best found perplexity while 3/12 of layers are using linear attention ($\lambda = 0.05$)

|  | Wikitext103 | | OpenWebText | | PG19 | |
|---|---|---|---|---|---|---|
| $\lambda$ | Best ↓ | $\Delta$ vs. base | Best ↓ | $\Delta$ vs. base | Best ↓ | $\Delta$ vs. base |
| – | 2.9400 | 0.0000 | 3.3200 | 0.0000 | 3.2000 | 0.0000 |
| 0.05 | 2.9300 | -0.0100 | 3.3606 | 0.0406 | 3.0300 | -0.1700 |
| 0.10 | 2.9509 | 0.0109 | 3.3770 | 0.0570 | 3.0526 | -0.1474 |
| 0.25 | 2.9554 | 0.0154 | 3.4175 | 0.0975 | 3.0661 | -0.1339 |
| 0.40 | 2.9958 | 0.0558 | 3.4402 | 0.1202 | 3.0858 | -0.1142 |
| 0.55 | 3.0050 | 0.0650 | 3.4523 | 0.1323 | 3.1141 | -0.0859 |

Table 1: Best validation cross-entropy loss (lower is better) and incremental change relative to the no-penalty baseline, across datasets and penalty $\lambda$. Green indicates improvement versus the respective dataset's baseline; red indicates degradation.

|  | Transformer | Performer | Reformer | AFT | (1+ELU) | Hedgehog | Ours ($\lambda$=0.05) | Ours ($\lambda$=0.10) | Ours ($\lambda$=0.25) |
|---|---|---|---|---|---|---|---|---|---|
| **Perplexity** | **18.6** | 26.8 | 25.6 | 28.2 | 25.6 | 20.8 | 18.7 | 19.1 | 19.2 |
| **CE loss (nats)** | **2.923** | 3.288 | 3.243 | 3.339 | 3.243 | 3.035 | 2.929 | 2.950 | 2.955 |

Table 2: Perplexity and cross-entropy (natural log of perplexity) on Wikitext-103 for GPT-2 style transformers and linear attention LLMs: Performer (Choromanski et al., 2021), Reformer (Kitaev et al., 2020), AFT (Zhai et al., 2021), and Hedgehog (Zhang et al., 2024). The other works' results are taken from Zhang et al. (2024).

## 6.2 MEMORY USAGE

Layer-wise hybrid LLM is more memory efficient because unlike original transformers, some of its layers have $\mathcal{O}(1)$ memory growth. Therefore, as the sequence lenghts becomes larger and larger, the gap between a hybrid LLM and an original Transformer will be more and more clear. Figure 1 shows the relative peak memory usage for transformer caches for each model obtained by our training method. The baseline is the original GPT-2 style transformer. The models trained on PG19 achieve a KV-cache cut ranging from 40 to 60% while all of them perform better in terms of perplexity. As an another example, OpenWebText models achieve a KV-cache cut ranging from 20 to 55 % while the most memory efficient one has a 14% worse perplexity. The model trained on Wikitext-103 using $\lambda = 0.25$ cuts the kv-cache usage by more than 40% while increasing perplexity by 1%. The saved memory can be utilized to increase the batch size (number of users effectively). Figure 2 shows the peak total memory usage of 2 of the hybrid models vs. number of tokens generated. Their memory usage is compared to full softmax LLM. The $\lambda = 0.1$ replace 6 while $\lambda = 0.25$ replaces 8 layers with linear attention.

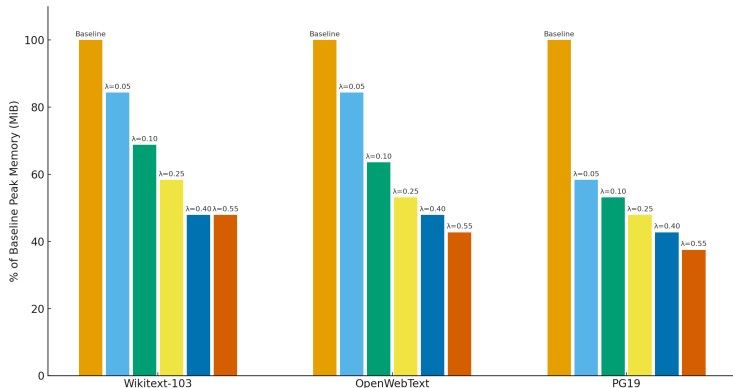

Figure 1: Peak Total Transformer Cache Memory Usage for Models Trained on Different Dataset with Different $\lambda$ values for generating 1024 tokens

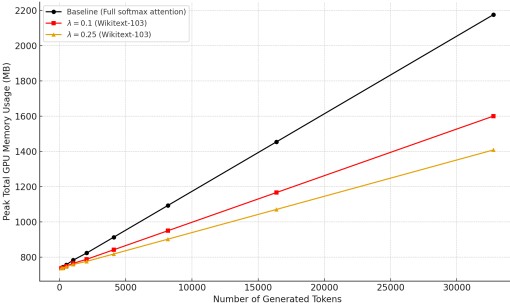

Figure 2: Peak total transformer memory usage vs. number of generated Tokens (two of the models are shown compared to the baseline)

| Dataset | $\lambda$ | T1 | T2 | T3 | T4 | T5 | T6 | T7 | T8 | T9 | T10 | T11 | T12 |
|---|---|---|---|---|---|---|---|---|---|---|---|---|---|
| Baseline | – | × | × | × | × | × | × | × | × | × | × | × | × |
| Wikitext-103 | 0.05 | o | o | × | × | × | × | × | × | × | × | × | o |
| | 0.10 | o | o | × | × | o | × | × | × | × | o | o | o |
| | 0.25 | o | o | o | o | o | × | × | × | o | o | o | o |
| | 0.40 | o | o | o | o | o | × | o | × | o | o | o | o |
| | 0.55 | o | o | o | o | o | o | × | × | o | o | o | o |
| OpenWebText | 0.05 | × | o | o | o | × | × | × | × | × | × | × | × |
| | 0.10 | × | o | o | o | o | o | o | × | × | × | o | o |
| | 0.25 | × | o | o | o | o | o | o | o | × | × | o | o |
| | 0.40 | × | o | o | o | o | o | o | o | o | × | o | o |
| | 0.55 | × | o | o | o | o | o | o | o | o | o | o | o |
| PG19 | 0.05 | × | o | o | o | o | o | o | × | × | × | o | o |
| | 0.10 | × | o | o | o | o | o | o | o | × | × | o | o |
| | 0.25 | × | o | o | o | o | o | o | o | × | o | o | o |
| | 0.40 | × | o | o | o | o | o | o | o | o | o | o | o |
| | 0.55 | o | o | o | o | o | o | o | o | o | o | o | o |

Table 3: The optimal layer configuration found by the proposed training algorithm

## 7 DISCUSSION

The pattern on the layers which have been replaced by linear attention has shown to be fairly consistent. Table 3 shows the configuration of replaced layers in each benchmark. Figure 3 shows the trajectory of $s^{(j)}$. The overall trend shows that at the beginning of training, all layers tend to linear since linear attention alone can significantly reduce the loss during the initial iterations. But as training goes on, some of the layers are required to switch back to softmax attention to make the model capable of reducing the loss further. Another observation is that in some cases introduction of linear layers and empowering the training to choose linear layer to replace softmax, not only doesn't

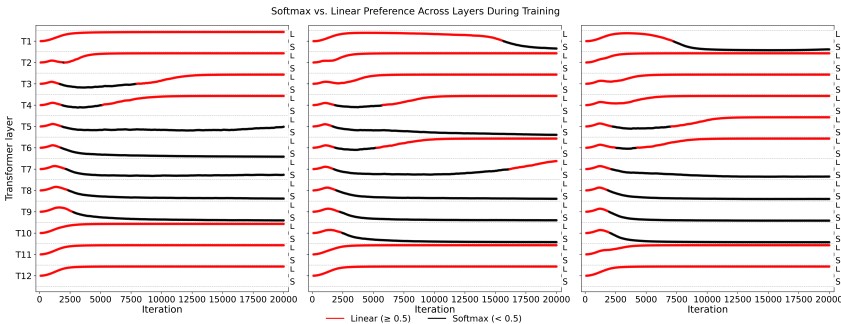

Figure 3: Layer Configuration During Training for $\lambda = 0.25$ Across Wikitext-103, PG19, and OpenWebText

lead to loss of accuracy but also it improves the best loss found with the original configuration. It is obvious that solutions reachable by our method are a superset of any all-softmax (traditional) or all-linear setting. Linear attention adds to the diversity of functions that a the model can approximate. A common trend is that earlier layers as well as final layers have a higher tendency to be replaced by linear attention while the layers in the middle need to stay as softmax attention.

We thank the reviewers for their thoughtful comments and valuable suggestions. In response, we have begun fine-tuning a Llama 3.2 3B model using our proposed method and evaluating it on a range of downstream tasks from LM-Eval-Harness (Biderman et al., 2024), including comparisons against random and structured linear-layer allocations. However, due to limited time, these experiments are not yet complete. We therefore require additional time to thoroughly conclude this study and plan to report the full set of results as future work building on this submission.

## 8 CONCLUSION

In this work, we presented a training-time method for constructing layer-based hybrid transformers. This optimization approach enables us to avoid brute-force exhaustive search in the extremely large space of possible configurations of hybrid models. We also show that the optimal (layer, attention type) configuration depends on the dataset and the maximum tolerable loss that the user is willing to accept, thus making it more practical to use a training-time optimization-based selection rather than exhaustive search.

We evaluated our method and showed that for a GPT-2 style model, we can achieve up to 40% reduction in cache memory usage, while increasing the perplexity only by 1%. We also show that, in some cases, the hybrid models obtained via the proposed method achieve significantly better perplexity.

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

# A APPENDIX

## A.1 TRAINING HYPERPARAMETERS

We used a 125M GPT-2 Style Transformer model (Radford et al., 2019). The following table includes the hyperparameter used for training.

| Category | Hyperparameter | Value |
|---|---|---|
| Training setup | Micro-batch size | 16 |
| | Block size (context length) | 1024 |
| Model architecture | Number of layers ($L$) | 12 |
| | Attention heads ($n_{\text{head}}$) | 12 |
| | Embedding dimension ($d$) | 768 |
| | FFN Dimension | 3072 |
| Optimizer (AdamW) | Learning rate (max) | 6e-4 |
| | Weight decay | 1e-2 |
| | $\beta_1$ | 0.9 |
| | $\beta_2$ | 0.95 |
| | Gradient clipping | 1.0 |
| Learning rate schedule | Decay learning rate | cosine |
| | Warmup steps | 2000 |
| | LR decay steps | 600,000 |
| | Minimum learning rate | 5e-5 |

Table 4: Training hyperparameters used in our experiments.

## A.2 PROOF OF THEOREM 3

*Proof.* By Lemma 1, the auxiliary term equals its soft expectation, so $\nabla \bar{J}_\tau$ is well-defined and smooth in $(\phi, u)$. Write the update as

$$\theta_{t+1} = \theta_t - \eta_t \Big( \nabla \bar{J}_{\tau_t}(\theta_t) + e_t + M_{t+1} + \Delta_t \Big), \qquad \theta = (\phi, s),$$

where $M_{t+1} := \widehat{\nabla} \bar{J}_{\tau_t}(\theta_t) - \mathbb{E}[\widehat{\nabla} \bar{J}_{\tau_t}(\theta_t) \mid \theta_t]$ is a martingale difference with bounded second moment by **A3**, $e_t := \mathbb{E}[\widehat{\nabla} \bar{J}_{\tau_t}(\theta_t) \mid \theta_t] - \nabla \bar{J}_{\tau_t}(\theta_t)$ is the ST bias, and $\Delta_t := \nabla \bar{J}_{\tau_{t+1}}(\theta_t) - \nabla \bar{J}_{\tau_t}(\theta_t)$ is the drift due to changing $\tau$. By **A4**, the conditional mean of the update is a *pseudogradient* uniformly aligned with $-\nabla \bar{J}_{\tau_t}$; by Lipschitzness in **A3**, $\|\Delta_t\| = O(|\tau_{t+1} - \tau_t|)$, so $\sum_t \eta_t \|\Delta_t\| < \infty$ by **A2**. The tanh bound (**A5**) yields compact sublevel sets, hence tightness of $\{\theta_t\}$. By **A4**, the conditional mean update is a pseudogradient uniformly aligned with $-\nabla \bar{J}_{\tau_t}$; this is the standard "gradient with errors/pseudogradient" setting Bertsekas & Tsitsiklis (2000); Borkar (2008); Kushner & Yin (2003). By **A3**, $\nabla \bar{J}_\tau$ is locally Lipschitz, hence $\|\Delta_t\| = \|\nabla \bar{J}_{\tau_{t+1}}(\theta_t) - \nabla \bar{J}_{\tau_t}(\theta_t)\| = O(|\tau_{t+1} - \tau_t|)$. Because $\sum_t \eta_t |\tau_{t+1} - \tau_t| < \infty$ and $\eta_t / \tau_t \to 0$ (**A2**), the temperature is a slowly varying parameter on a slower time scale Borkar (1997); Kushner & Yin (2003). The ODE/dynamical-systems method for stochastic approximation with martingale noise Kushner & Yin (2003); Borkar (2008); Benaim (1996) then implies that the limit set is a.s. internally chain transitive for the nonautonomous flow $\dot{\theta} = -\nabla \bar{J}_{\tau(t)}(\theta)$ and, by upper semicontinuity of stationary sets, is contained in $\lim_{\tau \downarrow 0} \Theta_\tau^\star$, proving (i). For (iii), minimizing $\bar{J}_\tau$ adds the uniform penalty $\frac{\lambda}{L}$ only to choosing $s$; as $\tau \to 0$, the gate deterministically picks the block minimizing $\text{CE}_j^i + \frac{\lambda}{L} \mathbf{1}\{j = s\}$, which is equivalent to the stated threshold on $\Delta \text{CE}_i$. $\qquad\square$

# B LLM USAGE

LLM were used in the process of writing of this paper to polish writing, grammar, and choice of words.

