# OpenReview forum: "Training-time Selection of Linear Vs. Softmax Attention in Layer-based Hybrid Transformers"
_ICLR.cc/2026/Conference — ICLR 2026 Conference Withdrawn Submission_

### Official Review · Reviewer_kJkW · 2025-10-27

**Soundness:** 2
**Presentation:** 1
**Contribution:** 2
**Rating:** 2
**Confidence:** 3

**Summary:**

The authors propose training full-softmax and linear attention in parallel and using a Gumbel-softmax-based differentiable binary gate to select between them. They regularize the selection to prefer linear attention, allowing the automatic selection of layers that the network strongly prefers to be full attention. This reduces memory usage and should improve inference speed, albeit at the cost of a more expensive training.

**Strengths:**

- Simple method
- Seems to work well
- Reduces memory usage
- Analysis in Tab. 3 and Fig 4.

**Weaknesses:**

- I think the writing quality is unacceptable for NeurIPS. For example, the method is described twice, redundantly: once in Sec. 4, then again in Sec. 5.
- The authors use the smallest, 125M param GPT-2 architecture for their experiments. While I am generally opposed to pushing for ever-larger models in academia, I think this is below the threshold of what is convincing in 2025. With a 4xH100 setup, one can relatively easily train a ~0.5B model, preferably on some proportion of OpenWebText.
- Small perplexity differences might result in relatively large downstream performance differences. Checking the performance on some zero-shot downstream tasks (e.g. BLiMP, CBT, PIQA) would be very useful.
- The authors should report some metric on how much this increases the cost of training by running both the softmax attention while training, for example, the number of flops required.
- The model discovers a relatively consistent pattern of attention allocation. Maybe this could be leveraged to set the attention types manually and avoid the training cost. Alternatively, maybe a short training run can be used to determine the attention types, and then a bigger run could use only one attention type at a time.

**Questions:**

- L043: causal mask missing (necessary during prefill and training)
- Eq. (1) causal mask not needed for inference: the model generates one token at a time in a loop, it can't see it's own future outputs
- L80: In complexity analysis, why do the authors consider the perfill not to be part of the time used by the model?
- L108: typo: vastyl
- Eq 5, Eq 8: \phi so far was used as a nonlinearity in linear attention (L123), I assume here it means parameters?
- Algorithm 1: skip connections missing
- L469: typo: our method our

---

> ### Author Response · Authors · 2025-11-21
>
> Thank you very much for your comments and suggestions. We intend to address all of your points in the revised version, which will be submitted by the final deadline.
>
> 1. We will revise the manuscript and resolve all the writing issues.
>
> 2. To address this point which was also raised by other reviewers, we will use a pretrained Llama 3.2 3B and finetune it with LoRA.
> 3. We’ll use the same base model to fine-tune hybrid models with random and structured hybrid allocations, and compare our model against them on a variety of benchmarks in LM-Eval-Harness.
>
> 4. We will include the metric and the training cost associated with our method in the revised submission.
>
> 5. Exactly as you pointed out, in our experiments with GPT-2 (125M), we generally observe that the outer layers are assigned linear attention. However, we also see counterexamples where this layer allocation pattern is not achieved. To examine whether this behavior persists at a larger scale, we are currently fine-tuning a Llama 3.2 3B model and will include these results in the revised version.
>
> We also appreciate your pointing out the writing errors in the manuscript.
>
> In complexity analysis, the prefill time is included as the first term of the sum, where the first output token is produced (M+1).
> \phi in eq. 5 and eq. 8 denotes the parameters, as you pointed out. In the revised version, we'll make sure that the naming will be consistent throughout the manuscript.

---

> > ### Comment · Reviewer_kJkW · 2025-11-24
> >
> > I want to thank the authors for their reply.
> >
> > The changes proposed point in the right direction. However, we do not yet know the results of those experiments, so it is not possible to judge whether the method will achieve the same performance gains or eventually fail. Thus, I am maintaining my score.

---

### Official Review · Reviewer_dJgP · 2025-10-27

**Soundness:** 1
**Presentation:** 1
**Contribution:** 1
**Rating:** 0
**Confidence:** 5

**Summary:**

This work proposes a hybrid model of softmax and linear attention layers in transformer language models, that trains the discrete choice of softmax vs. linear attention in each layer through Gumbel softmax. The proposed method is evaluated on 12-layer GPT-2 models using three datasets: Wikitext103, OpenWebText, and PG19.

**Strengths:**

The hybridization of linear and standard attention mechanisms is a timely and relevant topic.

**Weaknesses:**

The quality of this submission is far below acceptable standards on all fronts:

* Method: The proposed method is both incremental and unconvincing. It introduces additional engineering overhead---specifically, the use of Gumbel-Softmax and an extra loss term with a tunable hyperparameter $\lambda$ to encourage the use of linear attention---while requiring both the standard and linear attention to be performed in each layer during training. For these added complexities to be justified, there should be a clear and significant advantage over a simpler baseline (e.g., manually setting certain layers to use linear attention). However, such compelling results or motivation are not presented in this work.

This concern is compounded by the fact that multiple training runs are needed to tune $\lambda$; it is unclear how that could be more efficient than training models with different numbers of linear-attention layers directly with some educated guess and efficiency budget.

* Experiments: the experiments are conducted only for the 12-layer GPT-2 and only evaluated in terms of perplexity. Given the empirical/engineering nature of the work, this is insufficient. Even in academic settings, it is now common to train models with up to 1B parameters on roughly 10B tokens with *multiple downstream applications*, to meaningfully evaluate modern language modeling methods (see, e.g., [Ref1] below).

* Presentation: The writing quality is generally poor. Excessive space is devoted to superficial contents. Figures are of low resolution (e.g., Figures 1 and 4 are blurry), while Figures 2 and 3 are unnecessarily large and contain little information. Table 1 and Table 3 are mostly devoted to ablations on $\lambda$ (again consuming excessive space). Table 3’s first column is labeled “T1.00.” Section 2 contains only two short paragraphs and a single sentence. ETC.

Overall, the submission does not meet the standards required for publication at any major machine learning conference.

-----------------
Other comments:

> Linear attention mechanisms (Peng et al., 2021) offer O(N) time and constant memory complexity but fall short in language
modeling quality as measured by perplexity.

This statement is no longer accurate. Linear attention systems have achieved competitive performance in recent work (see, e.g., Yang et al., 2024 [Ref1]). More generally, the paper fails to take into account recent advances in linear attention (see more references cited in [Ref1])

[Ref1]: Yang, Kautz, Hatamizadeh. Gated Delta Networks: Improving Mamba2 with Delta Rule. ICLR 2025

**Questions:**

The reviewer has no further questions and considers it unlikely that this work will become acceptable after any rebuttal or discussion.

---

> ### Author Response · Authors · 2025-11-21
>
> We are grateful to you for your comments. We plan to address the points you have raised under ‘weaknesses’ :
>
> 1. Thanks to your comment and another reviewer's comment, in the revised version of the manuscript, we will emphasize that lambda is not a hyperparameter but rather a user-provided constraint specifying the tolerable additional loss beyond the loss achieved with the conventional softmax attention across all layers. This allows our model to replace as many layers as possible with linear attention while satisfying this constraint.
>
> 2. We will use a pretrained Llama 3.2 3B and finetune it with LoRA.  We will use the same base model to fine-tune hybrid models with both random and structured hybrid allocations, and compare our model against them on a variety of benchmarks in LM-Eval-Harness.
> 3. Thank you very much for highlighting the presentation issues. In the revised version, which will be uploaded by the final deadline, those issues will have been resolved.
>
> As to your “Other Comment”, we thank you for this observation and agree that our original wording was too strong. Recent work, including [Ref1] and other modern linear-attention variants, shows that linear-time models can achieve perplexity competitive with standard Transformers on several benchmarks. We will revise the introduction to soften this claim and explicitly acknowledge these advances, updating the related-work discussion to include [Ref1] and additional recent linear-attention papers.
> At the same time, our contribution is orthogonal to the specific linear mechanism: our method treats the linear block as a drop-in replacement and automatically learns a hybrid allocation of linear vs. softmax layers under a given resource budget. If a particular linear attention variant were uniformly as strong as softmax in our setting, our procedure would simply assign more (or all) layers to it. In the experiments we report, however, the best allocations found by our algorithm remain hybrid and consistently outperform all-linear baselines under the same constraints. We will clarify this point in the revised version.
> In long-context settings, full-attention Transformers are often trained with a shorter context window than that used at evaluation, so their underperformance largely reflects limited extrapolation rather than an intrinsic disadvantage of softmax attention. Our hybrid allocation method is tailored to this setting. Given a budget that allows full attention only in some layers, it automatically decides where to apply softmax attention and where to use linear attention.

---

> > ### Comment · Reviewer_dJgP · 2025-11-25
> >
> > I thank the authors for their response.
> >
> > However, the response does not address my original concerns. Superficially renaming "lambda" from "hyperparameter" to "user-provided constraint" does not resolve the core limitation that it must still be expensively tuned for each model. The writing issues cannot be resolved through minor revisions here---much of the superficial content (e.g., large tables) would need to be replaced with genuinely useful material, which is not straightforward here given the incremental nature of the proposed method.
> >
> > Therefore, I will maintain my score. In my assessment, the overall quality of the submission falls substantially below ICLR standards.

---

### Official Review · Reviewer_bzA3 · 2025-10-28

**Soundness:** 3
**Presentation:** 3
**Contribution:** 2
**Rating:** 4
**Confidence:** 4

**Summary:**

The paper proposes a training-time method for hybrid attention models (softmax and linear attention) that automatically decides, layer by layer, which one to use. The approach relies on a differentiable gating mechanism with Gumbel-Softmax hard selection, combined with a penalty term λ that trades off accuracy versus efficiency. The method discovers efficient hybrid architectures that reduce memory usage (up to 40% less KV-cache) while maintaining perplexity close to a full softmax baseline.

**Strengths:**

- **Clear problem motivation**: The quadratic cost of softmax attention and the trade-off between linear and softmax are well articulated.
- **Empirical benefits**: Demonstrated memory reductions of 20–60% with minimal or no accuracy loss are practically relevant for scaling LLMs.
- **Layer-wise insights**: The finding that early and final layers are more replaceable by linear attention while mid-layers require softmax could inspire further mechanistic or interpretability investigations.

**Weaknesses:**

- **Limited baselines**: The paper would benefit from simpler baselines e.g., manually fixing the same ratio as found by the model of the layers as linear and the rest as softmax, to demonstrate whether more granular information than ratio of layers is even influencing the result.
- **Lack of related work discussion**: The paper does not cite prior research on **layer/module selection with Gumbel-Softmax** (e.g., adaptive depth models [1], user-adaptive layer skipping [2]), nor the extensive line of **neural architecture search** using differentiable operator selection via Gumbel-Softmax [3].
- **Hyperparameter burden**: The λ parameter is difficult to set in practice and adds to the hyperparameter search space.
- **Writing and polish**: The draft contains multiple spelling/grammar issues (“that a the model”, “by our method our a superset”), which detract from readability.
- **Missing introduction**: The current introduction functions more like a background on self-attention rather than situating the contribution in the broader literature. This weakens positioning.
- **Lack of interpretability analysis**: The interesting discovered layer configurations (linear layers at edges, softmax in the middle) are not probed with mechanistic or probing methods that could substantiate the intuition.

[1] - Learning to Skip for Language Modeling

[2] - A User-Adaptive Layer Selection Framework for Very Deep Sequential Recommender Models

[3] - Differentiable Architecture Search with Ensemble Gumbel-Softmax

**Questions:**

Could mechanistic interpretability tools (e.g., attention pattern visualization, probing tasks) reveal why early and final layers are more replaceable?

---

> ### Author Response · Authors · 2025-11-21
>
> We appreciate your comments and suggestions, and we intend to address all the points you have raised under 'weaknesses' as outlined below.
>
> 1. We will use a pretrained Llama 3.2 3B and finetune it with LoRA.  We will use the same base model to fine-tune hybrid models with both random and structured hybrid allocations, and compare our model against them on a variety of benchmarks in LM-Eval-Harness.
>
> 2. Thank you for pointing out and suggesting the related works. The revised version will include all three references and their related discussion.
>
> 3. Thanks to your comment and another reviewer's comment, in the revised version of the manuscript, we will emphasize that lambda is not a hyperparameter but rather a user-provided constraint specifying the tolerable additional loss beyond the loss achieved with the conventional softmax attention across all layers. This allows our model to replace as many layers as possible with linear attention while satisfying this constraint.
>
> 4. Thank you for pointing out the grammar/spelling issues. We will revise the paper to correct all the grammar/spelling issues.
>
> 5. Thanks to you and another reviewer who pointed out a similar issue, we will focus more on the related works in the field of linear and hybrid attention in the introduction section to highlight our contribution and positioning.
>
> 6. We acknowledge the importance of the interpretability issue and plan to address it in future work. In this manuscript, we will include a statement reflecting this intention.

---

### Official Review · Reviewer_YoxD · 2025-11-01

**Soundness:** 2
**Presentation:** 2
**Contribution:** 2
**Rating:** 4
**Confidence:** 3

**Summary:**

This paper proposes a training-time method to construct layer-wise hybrid Transformers that choose between standard softmax attention and linear attention. The method introduces a per-layer learnable logit whose sign, sampled via a straight-through Gumbel-Softmax gate with an annealed temperature, selects one of the two attention blocks during training. A penalty proportional to the number of softmax-selected layers (scaled by $\lambda/L$) trades accuracy against efficiency; in the zero-temperature limit, the method aims to flip a layer to linear attention if the per-layer cross-entropy increase is below $\lambda/L$. After training, layers with positive logits are replaced by linear attention for inference. Experiments on GPT-2–style models (12 layers) trained on Wikitext-103, OpenWebText, and PG19 report up to ~40% KV-cache memory reduction with small perplexity increases (~1%), and in some cases small perplexity improvements versus an all-softmax baseline. The paper observes a consistent pattern where early and final layers tend to switch to linear attention, while middle layers remain softmax.

**Strengths:**

- **Clear and simple training-time mechanism**: the per-layer Straight-through Gumbel-Softmax gating is a nice way to determine how a Transformer's layers should be hybridized, with a cross-entropy threshold that makes sense. This provides a new take versus prior setups that merely hybridize in a fixed / uniform manner (e.g., every N layers)
  - This also provides a principled knob (λ) to trade off accuracy against memory, with an annealing schedule driving discrete selections.
- **Supportive empirics** Results on three datasets show plausible memory savings (up to 40–60% KV-cache reduction) with small perplexity degradation (1%) in some settings.
- **Interesting findings with layer placement**: the paper shows interesting results w.r.t. trends in which layers tend to switch to linear or softmax attention

**Weaknesses:**

- **Limited model scale and evaluation benchmarks**: The paper only reports results on GPT-2-style 12-layer Transformers on language modeling benchmarks. Even with just these trained checkpoints, it would be interesting to see the performance on a wider range of tasks, such as LM Eval or retrieval synthetics (MQAR, Passkey Retrieval / Needle-in-a-haystack) that are popular among softmax attention vs linear attention works.

- **Limited comparison** The primary comparison ablates $\lambda$ and compares with either *all* softmax attention or linear attention layers. But as a a hybridizing method, it would be good to compare performance to other mixed layer strategies, such as interspersing 1 softmax attention layer between N linear attention layers.

**Questions:**

Apart from the additional comparisons requested in the weaknesses section, I'm curious about the following:

1. Did you consider applying ST-Gumbel Softmax gating for hybridizing *pretrained* LLMs (e.g., Llama 3.2 3B or Qwen3 8B)? With the stated compute (4 H100s) (or even with just LoRA), you should be able to fine-tune these models and still compute the metrics needed to determine whether to make a layer a linear attention layer. The benefits + motivation of more informed linear attention placement would still hold, but the work could have even larger contribution as it now applies to modern LLMs and opens up the tasks you can evaluate on.

2. Restating the point on comparison above, how does the learned configuration compare against simple heuristics (e.g., make the first k and last k layers linear) or a small grid/random search over layer subsets?

---

> ### Author Response · Authors · 2025-11-21
>
> We appreciate your comments and suggestions, and we intend to incorporate them fully into our revised manuscript, which we plan to upload before the December 3rd deadline. Here are specific actions on the two comments you have provided.
>
> 1. Thanks to your suggestion, we will use a pretrained Llama 3.2 3B and finetune it with LoRA.
>
> 2. We’ll use the same base model to fine-tune hybrid models with random and structured hybrid allocations, and compare our model against them on a variety of benchmarks in LM-Eval-Harness.

---

### Note · Authors · 2025-12-03

**Comment:**

We thank the reviewers for their thoughtful comments and valuable suggestions. In response, we have begun fine-tuning a Llama 3.2 3B model using our proposed method and evaluating it on a range of downstream tasks from LM-Eval-Harness (Biderman et al., 2024), including comparisons against random and structured linear-layer allocations. However, due to limited time, these experiments are not yet complete. We therefore require additional time to thoroughly conclude this study and plan to report the full set of results as future work, building on this submission.

**Withdrawal Confirmation:**

I have read and agree with the venue's withdrawal policy on behalf of myself and my co-authors.